# Vision-Aided Localization and Mapping in Forested Environments Using Stereo Images

**DOI:** 10.3390/s23167043

**Published:** 2023-08-09

**Authors:** Lucas A. Wells, Woodam Chung

**Affiliations:** Department of Forest Engineering, Resources and Management, College of Forestry, Oregon State University, Corvallis, OR 97331, USA; woodam.chung@oregonstate.edu

**Keywords:** direct visual odometry, simultaneous localization and mapping, non-linear least squares, computer vision, GNSS/GPS

## Abstract

Forests are traditionally characterized by stand-level descriptors, such as basal area, mean diameter, and stem density. In recent years, there has been a growing interest in enhancing the resolution of forest inventory to examine the spatial structure and patterns of trees across landscapes. The spatial arrangement of individual trees is closely linked to various non-monetary forest aspects, including water quality, wildlife habitat, and aesthetics. Additionally, associating individual tree positions with dendrometric variables like diameter, taper, and species can provide data for highly optimized, site-specific silvicultural prescriptions designed to achieve diverse management objectives. Aerial photogrammetry has proven effective for mapping individual trees; however, its utility is limited due to the inability to directly estimate many dendrometric variables. In contrast, terrestrial mapping methods can directly observe essential individual tree characteristics, such as diameter, but their mapping accuracy is governed by the accuracy of the global satellite navigation system (GNSS) receiver and the density of the canopy obstructions between the receiver and the satellite constellation. In this paper, we introduce an integrated approach that combines a camera-based motion and tree detection system with GNSS positioning, yielding a stem map with twice the accuracy of using a consumer-grade GNSS receiver alone. We demonstrate that large-scale stem maps can be generated in real time, achieving a root mean squared position error of 2.16 m. We offer an in-depth explanation of a visual egomotion estimation algorithm designed to enhance the local consistency of GNSS-based positioning. Additionally, we present a least squares minimization technique for concurrently optimizing the pose track and the positions of individual tree stem[s].

## 1. Introduction

The widespread adoption of sustainable forestry practices in much of the developed world has given rise to increasingly complex forest management objectives catering to a diverse array of interests. Consequently, silvicultural prescriptions often take into account numerous non-monetary factors, including forest resiliency and adaptability, wildlife conservation, aesthetic preservation, hydrological values, and other ecological functions that are dependent on stand structure [1]. In general, the incorporation of non-timber values into management objectives has served as the primary driver for transitioning silvicultural practices from homogeneous, even-aged systems to heterogeneous, uneven-aged systems. This paradigm shift directly affects forest operations; implementing complex silvicultural prescriptions becomes more costly in terms of layout, harvesting, and administration [2]. This challenge has spurred the demand for advanced precision forestry tools that offer accurate, real-time machine positioning, as well as forest measurement and mapping capabilities. Such tools have the potential to eliminate certain layout tasks, such as individual tree and boundary marking, thereby reducing operational costs and enhancing the economic feasibility of alternative silvicultural systems.

Real-time machine positioning and forest mapping are essential components of sustainable forest harvesting [3]. However, practical low-cost technologies for precise localization and large-scale mapping in forested environments during harvesting remain underdeveloped. Unlike the substantial advancements in automation introduced in agricultural systems [4], forest harvesting has not experienced groundbreaking technological progress in terms of automation and robotics. This lag can be attributed, in part, to the challenging environments in which forest machines operate, as well as potential cultural factors, resistance to change, or inadequate research and development efforts.

In this paper, we introduce visual-processing algorithms that facilitate precise real-time machine positioning and forest mapping. We employ visual egomotion estimation from a stereo camera to maintain the position of a forestry machine during instances of degraded or failed reception from a global navigation satellite system (GNSS). Furthermore, we demonstrate how precise position maintenance can be utilized to generate tree maps when combined with a tree-detection algorithm. Below, we itemize the main contributions of this work:A novel camera-based system that integrates GNSS, visual odometry, and a tree-detection system and egomotion estimation.A low-cost system using an off-the-shelf stereo camera and a consumer-grade GNSS receiver.An efficient pixel-selection method with predictable inter-frame runtime for direct image alignment.A global orientation parameter in the optimization framework for preliminary alignment between the GNSS and visual odometry pose track.A system that provides the tree position and dendrometric information to the user in real time.

In the remainder of this introduction, we will discuss the crucial role of GNSS in forest operations, along with the associated challenges and limitations. Additionally, we will introduce the concepts of visual odometry (VO) and simultaneous localization and mapping (SLAM).

### 1.1. Global Positioning

Global navigation satellite systems (GNSSs) have played a significant role in forest resource management. These systems have been employed for inventory plot localization [5], forest traverse surveys [6], mapping forest disturbances [7], machine tracking [8,9], automated time studies [10,11], and general operational monitoring [12]. Recently, GNSS technology has been applied to enhance occupational safety in logging operations by providing virtual geofences around workers on the job site [13,14,15]. Most modern forest machines come equipped with a GNSS receiver from the factory, enabling real-time positioning of the machine during operation.

While highly accurate GNSS positioning may not be essential for many forestry applications, it becomes crucial when constraining machines with a virtual boundary, particularly when boundaries coincide with ownership demarcations or delineate high-value, sensitive, or hazardous areas. High-precision localization is also necessary for mapping individual trees. Challenges with GNSS accuracy have impeded the widespread adoption of GNSS technology in the virtual boundary and mapping domains.

GNSS accuracy depends on various factors, one of which is the number and geometry of visible satellites. This is characterized by an index called position dilution of precision (PDOP), which is a multiplicative term that scales the expected accuracy of the receiver. PDOP values less than 1 provide the highest possible accuracy, while PDOP values greater than 20 generally render coordinate readings futile, e.g., a PDOP value of 20 with a GNSS receiver capable of 3 m accuracy results in an actual accuracy of 60 m. The forest canopy can block signals from reaching the receiver, an effect known as the canopy effect [6]. Another important factor—primarily responsible for degraded GNSS accuracy under forest canopy—is signal diffraction and reflection, known as multipath errors [16]. Lastly, GNSS accuracy also depends on the class of receiver, such as survey-grade or consumer-grade receivers. Research on GNSS accuracy in forested environments has shown consumer-grade receivers to have an accuracy range between 4 and 12 m [17,18,19,20,21]. In this work, we utilize a specific GNSS called the global positioning system (GPS), operated by the United States Air Force. The GPS satellite constellation comprises 33 satellites, 31 of which are currently operational.

### 1.2. Visual Odometry

Estimating the egocentric motion of a camera is a fundamental task in computer vision. Egomotion estimation aims to determine a 3D geometric transformation that describes the incremental translational and rotational change of a camera in motion relative to the observed environment. Motion estimation occurs in discrete time steps, where a new estimate at time *t* reveals the camera’s motion relative to its pose at time t−1. Each new estimate can be incrementally composed with the previous to produce a pose graph, i.e., a track of position and orientation over time. Visual odometry (VO), coined by Nistér et al. [22], is the term often used to describe time-integrated motion estimates and is analogous to other variants of odometry, e.g., wheeled odometry, where wheel encoders are used to estimate the traveled distance in ground vehicles. Odometric navigation is subject to errors that accumulate over time, eventually leading to positional drift, a well-known property of dead reckoning navigation systems.

Early research on the problem of recovering relative camera poses tackled the problem of structure-from-motion [23,24]. Structure-from-motion is the problem of recovering both 3D structure and camera poses from a set of unordered images. VO is a special case of structure-from-motion, where images are sequenced and pose recovery is the sole objective. To date, most algorithms for VO are based on a feature extraction and matching pipeline [25,26,27,28]. These algorithms are known in the vision community as feature-based methods. In general, feature-based methods involve three steps: (1) feature extraction and description using one of many available approaches (e.g., FAST detector [29,30], SIFT detector/descriptor [31] Harris detector [32], Shi and Tomasi detector [33], and SURF detector/descriptor [34]); (2) temporal feature matching, e.g., using mutual consistency or constrained matching [35]; and (3) pose recovery by minimizing the 3D-2D reprojection error using a perspective from *n* point (PnP) algorithm, e.g., EPnP [36], which is typically embedded in a random sample consensus (RANSAC) scheme [37]. See [38] for an overview of VO fundamentals in the context of feature-based methods.

Recent approaches to VO have migrated away from feature-based approaches due to complexity and the plethora of configurations in feature-based pipelines. Recent approaches use image intensities directly rather than extracting and matching features [39,40,41,42,43,44]. These methods are called direct methods and are founded on the work by Lucas and Kanade [45] introducing parametric image alignment. In contrast to feature-based methods, which recover relative camera motion by minimizing reprojection error, direct methods minimize the photometric error, the sum of squared differences in image intensities between two consecutive frames.

In this work, we employ a direct method to solve for the incremental 6-DoF motion parameters. We provide a detailed description of image warping and pixel selection techniques, as well as the optimization procedure. Our method is somewhat simplified in contrast to state-of-the-art methods, e.g., [40,43], which involve keyframe selection and windowed bundle adjustment optimization to mitigate trajectory drift. As we will show in the localization and mapping section, we align the odometry track with global coordinates from a GNSS, and therefore, maintaining global consistency within our VO framework is unnecessary. Instead, in this work, we focus on local consistency and computational efficiency.

### 1.3. Simultaneous Localization and Mapping

SLAM is a critical problem in robotics, which involves constructing a map of an unknown environment while concurrently determining the robot’s location within it. If the robot’s position and orientation are known at any given time, mapping its surroundings using sensor data becomes a straightforward task. Conversely, if the map is already known, the problem is reduced to localization, where the robot’s observations are utilized to determine its position and orientation within the map. When neither the pose nor the map is known, both the map and path must be estimated simultaneously. SLAM can be broadly classified into online and full SLAM problems. Online SLAM incrementally estimates the current pose along with the map, while full SLAM estimates the entire path and map using data from all poses and observations [46]. There has been research in forest harvesting related to the localization problem [47,48]; however, a stem map is required in order to localize the machine, which we consider to be a major limitation of such an approach since stem maps, in general, are not readily available.

Visual odometry (VO) and vision-based SLAM are closely related, as VO provides an estimate of the robot’s path. The key difference is that SLAM mandates maintaining a map—even if it is not ultimately needed—so that the robot can recognize previously explored areas. When the robot revisits an area on the map, it provides additional constraints to optimize the trajectory and map, ensuring global consistency. This process is known as loop closure. Generally, VO is focused on local consistency, while SLAM aims for global consistency. It is worth noting that if VO is free from drift, the SLAM problem is reduced to merely mapping observations. However, error-free VO has never been achieved in practice, making loop closures essential for maintaining global consistency. To recognize loop closures, robots need to operate in environments containing distinct features that can be reidentified upon returning to a previously mapped location.

Detecting loop closures in forested environments can be challenging, as the spatial configurations of features, such as trees, may not be unique enough to provide robust information for redetection. Integrating GNSS positioning with SLAM can assist in identifying loop closures by maintaining a globally consistent position path. In this paper, we demonstrate how intermittent and degraded GPS reception can be fused with VO to provide a globally consistent position estimate. Additionally, we present a graph-based SLAM algorithm for refining the estimated path and map simultaneously. For an introductory tutorial on graph-based SLAM, readers can refer to [49].

### 1.4. Notation

In our method description below, we denote vectors as bold lowercase letters, e.g., v, matrices as bold capital letters, e.g., M, and scalars as lowercase italic letters, e.g., *s*. We use the notation ∥·∥ as a shorthand for ${∥\cdot ∥}_{2}$, i.e., the Euclidean norm. We represent images as functions, $I:\Omega \to R^{3}$ for 3-channel color images, and I:Ω→R for gray-scale images where $\Omega \subset R^{2}$ is the image domain. Sets are denoted by capital script letters, e.g., A, and the number of elements in a set is given by |A|. See Appendix A for an overview of Lie groups and rigid transformations used in this paper.

## 2. Direct Visual Odometry

Given a reference image $I_{t-1}:\Omega \to R$ acquired at time t−1 and an input image $I_{t}:\Omega \to R$ acquired at time *t*, we seek to estimate the 3D egomotion of the camera between the frames. Estimating the camera’s motion is performed by solving for the parameters of a *warp*, $\xi_{t-1:t}$, which relates the pixels in $I_{t-1}$ to the pixels in $I_{t}$. For brevity, we drop the time subscript on the warping parameters and denote it by ξ, and denote the reference image by I and the input image by $I^{\prime }$. The intensity of a pixel in the reference image is given by I(p), where $p={\left(u,v\right)}^{T}\in \Omega$. Similarly, the intensity of a pixel in the input image is given by $I^{\prime }\left(p^{\prime }\right)$, where the position vector $p^{\prime }$ is the result after warping p according to the parameters ξ,
(1)$$\begin{array}{c} p^{\prime }=W\left(p;\xi \right). \end{array}$$

Leaving the warping function undefined for the moment, the images are registered, or aligned, by minimizing the photometric error according to the following objective:(2)$$\begin{array}{c} \min_{\xi }\;\sum_{p\in \Phi }{∥\;I\left(p\right)-I^{\prime }\left(W(p;\xi )\right)∥}^{2}\;, \end{array}$$
where Φ⊂Ω is a set of selected pixels from the image domain. Solving this expression is non-linear regardless of the warping function, as pixel intensities are unrelated to their coordinates. A common method for minimizing the objective is the Gauss–Newton (GN) algorithm. In general, the expression
(3)$$\begin{array}{c} \sum_{p\in \Phi }{∥\;I\left(p\right)-I^{\prime }\left(W(p;\xi +\Delta \xi )\right)∥}^{2}\;, \end{array}$$
is linearized w.r.t. some small change in the parameters, Δξ, and then the parameters are updated by
(4)$$\begin{array}{c} \xi_{\kappa +1}=\xi_{\kappa }+\Delta \xi \;, \end{array}$$
until ξ and Δξ converge.

This formulation, as well as a solution procedure, was first given by Lucas and Kanade [45] in their seminal work, on which parametric image alignment, and subsequently direct VO, is founded. For a detailed presentation of optimization algorithms and alternative formulations, see Baker and Matthews [50]. We will discuss our optimization technique in a latter section after introducing the warping function.

### 2.1. Image Warping

The egomotion of a camera with no assumed holonomic constraints has 6 degrees of freedom; rotation about the x,y and *z* axes and translation along the x,y and *z* axes. The motion according to these degrees of freedom is represented by a transformation matrix in the special Euclidean Lie group:(5)$$\begin{array}{c} T=\left(\begin{array}{cc} R & t\\ 0^{T} & 1 \end{array}\right)\in \mathrm{SE}\left(3\right)\;, \end{array}$$
where R∈SO(3) is a 3D rotation matrix and $t\in R^{3}$ is a 3D displacement or translation vector. This is a rigid-body transformation that encodes the change in rotation and translation of a non-deformable object in motion between two discrete points in time. Although the matrix T has 6 degrees of freedom, there are 12 values that make up the non-homogeneous portion of the matrix: 9 values in the rotation matrix and 3 values for translation. This imposes unnecessary computational demands in an optimization setting. For this reason, we use the Lie algebra se(3) to parameterize the transformation. We denote the transformation matrix as a function of the parameters ξ∈se(3),
(6)$$\begin{array}{c} T\left(\xi \right)=\exp\left(\hat{\xi }\right)\in \mathrm{SE}\left(3\right)\;. \end{array}$$The exponential map of se(3) can be computed in closed form, as can the logarithm map that takes SE(3) back to the algebra se(3),
(7)$$\begin{array}{c} \xi =\ln\left(T(\xi )\right)\in \mathrm{se}\left(3\right)\;. \end{array}$$Using the Lie algebra parameterization of the transformation, we define a warping function $W:\left(R^{3}\times R^{6}\right)\to R^{2}$ that takes a homogeneous pixel coordinate as an input, back projects it to $R^{3}$, transforms the back projection according to the warping parameters, and then projects it back to $R^{2}$,
(8)$$\begin{array}{c} W\left(\tilde{p};\xi \right)=\pi \left(T\left(\xi \right)\pi^{-1}\left(\tilde{p},z\right)\right)\;, \end{array}$$
where the notation $\tilde{p}={\left(u,v,1\right)}^{T}\in P^{2}$ denotes a pixel position vector in homogeneous coordinates. The function $\pi^{-1}\left(\cdot \right)$ performs the back-projection and is defined as
(9)$$\begin{array}{c} \pi^{-1}\left(\tilde{p},z\right)={\left({\left(zK^{-1}\tilde{p}\right)}^{T},1\right)}^{T}\in R^{4}\;, \end{array}$$
where *z* is a depth measurement for the pixel and $K^{-1}$ is the inverse of the projection matrix. Note that we homogenized, i.e., appended a fictitious coordinate, to the back-projected vector so that it is compatible with the transformation matrix. The transformation matrix is not homogeneous, i.e., the last row of the matrix shown in Equation (Equation 5) is omitted, so the resulting vector following the transformation has three dimensions. The function that performs the forward projection is defined as
(10)$$\begin{array}{c} \pi \left(x\right)=\tilde{n}\left(Kx\right)\in R^{2}\;, \end{array}$$
where $x={\left(x,y,z\right)}^{T}\in R^{3}$, K is the camera projection matrix, and $\tilde{n}:R^{3}\to R^{2}$ normalizes the homogeneous coordinates, i.e., $\tilde{n}\left({\left(x,y,w\right)}^{T}\right)={\left(x/w,y/w\right)}^{T}$. Finally, the projection matrix and its inverse are defined as
(11)$$K=\left(\begin{array}{ccc} f & 0 & c_{u}\\ 0 & f & c_{v}\\ 0 & 0 & 1 \end{array}\right)\;,\;K^{-1}=\left(\begin{array}{ccc} \frac{1}{f} & 0 & -\frac{c_{u}}{f}\\ 0 & \frac{1}{f} & -\frac{c_{v}}{f}\\ 0 & 0 & 1 \end{array}\right)\;,$$
where *f* is the focal length, assuming unit aspect ratio pixels, and $\left(c_{u},c_{v}\right)$ is the principal point.

The only variable that still needs attention is the depth measurement *z* used in the back-projection function. We estimate the depth of each pixel by first computing the stereo correspondence via semi-global matching [51]. We use a real-time GPU implementation presented in [52]. Due to the smoothness constraints imposed by the semi-global matching scheme, occlusions are filled by some non-zero value. We handle occlusions by right–left consistency check: The disparity map $D^{l,r}$ is computed first, then the second disparity map $D^{r,l}$ by reflecting the left and right images along the *v*-axis and using the right image in place of the left, and the left in place of the right. The final disparity map D is equal to $D^{l,r}$ when the absolute difference between $D^{l,r}$ and $D^{r,l}$, evaluated at a position vector p, does not exceed some threshold δ. Otherwise, D(p) takes zero,
(12)$$\begin{array}{c} D\left(p\right)=\left\{\begin{array}{cc} D^{l,r}\left(p\right) & \mathrm{if}\;|D^{l,r}\left(p\right)-D^{r,l}\left(p\right)|\leq \delta \\ 0 & \mathrm{otherwise} \end{array}\right.,\;\forall p\in \Omega \;. \end{array}$$In this work, we use δ=1. We do not perform sub-pixel refinement to interpolate disparity values. We simply use positive integers to represent the disparity map, D:Ω→N. The depth estimate, *z*, for the pixel p is given via triangulation,
(13)$$\begin{array}{c} z\left(p\right)=\frac{fb}{D(p)}\;, \end{array}$$
where *f* is the focal length and *b* is the baseline distance of the stereo rig in centimeters. This assumes that the stereo camera is calibrated and row-aligned. We follow methods presented in [53] for camera calibration.

### 2.2. Optimization

We will now extend our discussion regarding the minimization of photometric error. As noted earlier, minimizing the expression defined in Equation (Equation 2) is a non-linear optimization task. This formulation requires a linearization step during each GN iteration. Namely, the Jacobian of the warp and the Hessian need to be computed during each iteration, which can lead to significant computational demands depending on the size of the Jacobian. Following from Baker and Matthews [50], we redefine the objective under the inverse compositional (IC) formulation by interchanging the roles of the reference and input image and solving for incremental warp parameters instead of additive updates as in the Lucas–Kanade formulation. Given some initial guess of the parameters, ξ, the objective is to minimize the following expression w.r.t the incremental warp parameters,
(14)$$\begin{array}{c} \Delta \xi^{★}=\underset{\Delta \xi }{\mathrm{argmin}}\sum_{p\in \Phi }{∥\;I\left(W(\tilde{p};\Delta \xi )\right)-I^{\prime }\left(W\left(\tilde{p};\xi \right)\right)∥}^{2}\;. \end{array}$$

The parameters are updated by inverting the incremental warp parameters and composing with the current estimate, $\xi \leftarrow \xi \circ \Delta \xi^{-1}$, where the notation ∘ denotes composition. The incremental warp parameters need to be inverted at each GN iteration as the linearization, which we will discuss next, is performed on the reference image. The update rule can be explicitly written as
(15)$$\begin{array}{cc} \xi_{\kappa +1} & =\xi_{\kappa }\circ \Delta \xi^{-1}\\ \; & =\ln\left(\exp\left({\hat{\xi }}_{\kappa }\right)\exp\left(-\Delta \hat{\xi }\right)\right)\\ \; & =\ln\left(\begin{array}{cc} R_{\kappa }\Delta R^{T} & R_{\kappa }\left(-\Delta R^{T}\Delta t\right)+t_{\kappa }\\ 0^{T} & 1 \end{array}\right)\;. \end{array}$$According to the GN algorithm, the incremental warp parameters Δξ are given by the normal equations,
(16)$$\begin{array}{c} J^{T}J\Delta \xi =-J^{T}r\Rightarrow \Delta \xi =-{\left(J^{T}J\right)}^{-1}J^{T}r\;, \end{array}$$
where J is a m×6 Jacobian matrix, $J^{T}J$ is the Gauss–Newton approximation of the Hessian matrix, and r is the vector of residuals given by
(17)$$\begin{array}{c} r=I^{\prime }\left(W(\tilde{p};\xi )\right)-I\left(p\right)\;. \end{array}$$The Jacobian encodes the partial derivatives of the reference image at each pixel $p_{{\left\{i\right\}}_{1}^{m}}$ with respect to the six warping parameters,
(18)$$J=\left(\begin{array}{c} {\frac{\partial I(p_{1})}{\partial \xi }}^{T}\\ {\frac{\partial I(p_{2})}{\partial \xi }}^{T}\\ \vdots \\ {\frac{I(p_{m})}{\partial \xi }}^{T} \end{array}\right)=\left(\begin{array}{cccc} \frac{\partial I(p_{1})}{\partial \xi_{1}} & \frac{\partial I(p_{1})}{\partial \xi_{2}} & \ldots & \frac{\partial I(p_{1})}{\partial \xi_{6}}\\ \frac{\partial I(p_{2})}{\partial \xi_{1}} & \frac{\partial I(p_{2})}{\partial \xi_{2}} & \ldots & \frac{\partial I(p_{2})}{\partial \xi_{6}}\\ \vdots & \vdots & \ddots & \vdots \\ \frac{\partial I(p_{m})}{\partial \xi_{1}} & \frac{\partial I(p_{m})}{\partial \xi_{2}} & \ldots & \frac{\partial I(p_{m})}{\partial \xi_{6}} \end{array}\right)\;.$$The linearization of Equation (Equation 14) can be achieved by performing a first-order Taylor expansion about the current estimate of the parameters. Denoting the *i*th row in the Jacobian as ∂I(p)∂ξ, corresponding to some pixel p∈Φ and applying the chain rule, we obtain
(19)$$\begin{array}{c} {\frac{\partial I(p)}{\partial \xi }}^{T}=\frac{\partial I(p)}{\partial p}\frac{\partial p}{\partial x}\frac{\partial x}{\partial \xi }\;. \end{array}$$The partial derivative of the reference image w.r.t. some pixel position p is simply the gradient vector of the image along the *u* and *v* axes,
(20)$$\begin{array}{c} \frac{\partial I(p)}{\partial p}=\nabla I\left(p\right)=\left(I_{u}\left(p\right),I_{v}\left(p\right)\right)\;. \end{array}$$We purposefully denoted the gradient vector as a row vector. Taking p to be equal to a reduced form of the forward projection function π(·) such that p=fxz+cu,fyz+cvT, and $x={\left(x,y,z\right)}^{T}$ to be a back projection of the pixel, the partial derivative can be written as
(21)$$\begin{array}{c} \frac{\partial p}{\partial x}=\left(\begin{array}{ccc} \frac{f}{z} & 0 & -f\frac{x}{z^{2}}\\ 0 & \frac{f}{z} & -f\frac{y}{z^{2}} \end{array}\right)\;. \end{array}$$Finally, the partial derivative of the back-projected pixel x w.r.t. the warping parameters evaluated at the identity warp ξ=0 takes the form,
(22)$$\begin{array}{c} {\left.\frac{\partial x}{\partial \xi }\right|}_{\xi =0}=\left({\left[x\right]}_{\times }I_{3}\right)=\left(\begin{array}{cccccc} 0 & -z & y & 1 & 0 & 0\\ z & 0 & -x & 0 & 1 & 0\\ -y & x & 0 & 0 & 0 & 1 \end{array}\right)\;. \end{array}$$This result follows from the skew–symmetric matrix operator used when computing the SE(3) exponential map of the warp parameters. Multiplying out the last two partials gives
(23)$$\begin{array}{cc} {\frac{\partial I(p)}{\partial \xi }}^{T} & =\nabla I\left(p\right)\frac{\partial p}{\partial x}\frac{\partial x}{\partial \xi }\\ \; & =\left(I_{u}\left(p\right),I_{v}\left(p\right)\right)\left(\begin{array}{cccccc} f\frac{xy}{z^{2}} & f\frac{x^{2}-z^{2}}{z^{2}} & \frac{xy}{z} & \frac{f}{z} & 0 & -f\frac{x}{z^{2}}\\ f\frac{y^{2}+z^{2}}{z^{2}} & -f\frac{xy}{z^{2}} & -f\frac{x}{z} & 0 & \frac{f}{z} & -f\frac{y}{z^{2}} \end{array}\right)\;. \end{array}$$For convenience, we also show a row in the Jacobian corresponding to some pixel p written out explicitly,
(24)$$\begin{array}{c} \frac{\partial I(p)}{\partial \xi }={\left(\begin{array}{c} \frac{I_{u}\left(p\right)fxy+I_{v}\left(p\right)\left(fy^{2}+fz^{2}\right)}{z^{2}}\\ \frac{I_{u}\left(p\right)\left(-fx^{2}-fz^{2}\right)-I_{v}\left(p\right)fxy}{z^{2}}\\ \frac{I_{u}\left(p\right)fy-I_{v}\left(p\right)fx}{z}\\ \frac{I_{u}\left(p\right)f}{z}\\ \frac{I_{v}\left(p\right)f}{z}\\ \frac{-I_{u}\left(p\right)fx-I_{v}\left(p\right)fy}{z^{2}} \end{array}\right)}^{T}\in R^{6}\;. \end{array}$$The row vector stated above is computed for each pixel p∈Φ and stacked into the m×6 Jacobian matrix, again where *m* is the number of pixels in Φ. The columns of the Jacobian can be visualized as the steepest descent images shown in Figure 1.

Since the linearization is performed on the coordinate frame of the reference image, the Jacobian J and the GN approximation of the Hessian $H=J^{T}J$, as well as its inverse $H^{-1}$ only need to be computed once. These are the computational savings of the IC formulation [50] over the original Lucas–Kanade algorithm [45].

#### 2.2.1. Pixel Selection

Since minimizing photometric error is based on image gradients, only pixels with a non-zero gradient contribute to optimization. Therefore, including pixels with zero gradient in the Jacobian introduces unnecessary computations. In feature-rich environments, such as forests, selecting pixels with a non-zero gradient might not significantly reduce the number of pixels used in optimization. As shown in Figure 2a, selecting all non-zero gradient pixels results in using approximately 85% of the image. We can further reduce the number of pixels, thus decreasing computation, by thresholding pixels based on gradient magnitude. This approach, however, results in a Jacobian matrix that is subject to change in dimension between frames since the distribution of gradient magnitude is not guaranteed to be consistent; we must allocate enough memory to store the expected maximum number of non-zero gradient pixels across all frames, which cannot be determined in advance, or dynamically allocate memory prior to optimizing each frame. This issue can be resolved by selecting a percentage of the total number of pixels in the image either by performing binary search for a gradient magnitude threshold that results in the desired number of pixels or sorting the gradients in descending order and selecting the first $N\frac{p}{100}$ pixels, where *p* is the desired percentage and *N* is the number of pixels in the image. In Figure 2b–d, we show the selected pixels resulting from desired percentages ranging from 25% to 75%.

Prior to selecting pixels based on the gradient magnitude, we discard all pixels with a disparity value of zero since these pixels are projected to infinity during image warping. Zero disparity pixels are apparent in Figure 2 as areas in the images that obviously have non-zero gradients but are too distant to have a non-zero disparity value.

#### 2.2.2. Robustness

GN optimization assumes Gaussian distributed errors. It is often the case in real-world data, however, that non-Gaussian errors arise due to inaccurate pixel correspondences during disparity computation, motion-induced occlusions, illumination changes and auto-exposure adjustments. A cost function for minimizing photometric error that is insensitive to non-Gaussian distributed noise is said to be robust. In the GN framework, this can be achieved with iterative re-weighted least squares (IRLS) using a robust cost function. The decision of the cost function is somewhat arbitrary, typically selected through empirical evaluation or by means of some prior knowledge regarding the structure of outliers in the data. In this work, we choose to use Tukey’s biweight cost function [54,55] since it suppresses large residuals in contrast to Huber’s cost function [56] which simply down weights their influence.

Tukey’s biweight cost function takes the form
(25)$$\begin{array}{c} \rho \left(r_{i}\right)=\left\{\begin{array}{cc} {\left(1-{\left(\frac{r_{i}}{c}\right)}^{2}\right)}^{2} & \mathrm{if}\;|r_{i}|\leq c\\ 0 & \mathrm{otherwise} \end{array}\right.\;, \end{array}$$
where the constant *c* is usually chosen to be 4.6851 to achieve 95% asymptotic efficiency with the normal distribution and $r_{i}$ is the *i*th residual from the error vector given by Equation (Equation 17). The cost function assumes that the residuals have unit variance. A common choice to estimate the scale parameter to standardize the residuals is to compute the median absolute deviation (MAD) and multiply by the expected MAD for a standard normal distribution,
(26)$$\begin{array}{c} \hat{s}=k\cdot {\mathrm{median}}_{i}\;\left|r_{i}\right|\;, \end{array}$$
where k=1.4826. To incorporate robustness in the GN minimization routine, we construct a weight vector w, where each weight wi=ρ(ris^) and set the diagonal entries of a weight matrix equal to the weight vector, i.e., W=diag(w). It follows that the solution to the incremental update of the linearized system under the IRLS framework takes the form
(27)$$\begin{array}{c} \Delta \xi =-{\left(J^{T}WJ\right)}^{-1}J^{T}Wr\;, \end{array}$$
which is used in place of the normal equations shown in Equation (Equation 16).

## 3. Localization and Mapping

In this section, we present a graph-based algorithm for maintaining a globally and locally consistent pose track using the estimated frame-to-frame egomotion parameters from the previous section and a consumer-grade GPS receiver. We also show how the optimized pose graph can be refined to generate a map of detected tree stems. Our algorithm consists of two phases: global alignment, in which we minimize errors between the odometry-based pose graph and global positions provided by the GPS receiver, and local refinement, where we relate poses by multiple observations of tree stems and simultaneously optimize the configuration of the pose graph and tree stem positions.

In order to increase the efficiency of computations, we use the se(2) Lie algebra of rigid transformations to represent the graph, and ultimately the map, as opposed to the se(3) parameters we optimized for in the odometry section. We do this by simply extracting the translation parameters corresponding to the *x* and *z* axes in se(3) to represent the translation along the *x* and *y* saxes on a planar pose graph, and the rotation component about the *y* axis from the se(3) parameters to represent the heading. Using se(2) parameters produces a meaningful map that can easily be presented on a 2D display monitor.

### 3.1. Global Alignment

Given a set of frame-to-frame odometry observations, $\left\{\Delta \xi_{1},\Delta \xi_{2},\ldots ,\Delta \xi_{m}\right\}$, where each $\Delta \xi_{i}={\left(x,y,\theta \right)}^{T}\in \mathrm{se}\left(2\right)$, we seek to align a pose graph constructed from the odometry observations with a set of global position readings from a GPS receiver. We assume odometry observations to be locally consistent but subject to drift and assume GPS coordinates to be locally bounded by a Gaussian distribution specified by an arbitrarily large covariance matrix that captures the expected errors due to multi-pass signals and geometric dillution of precision. We denote a GPS coordinate as $g={\left(x,y\right)}^{T}\in R^{2}$, where *x* and *y* represent the global position estimate in meters within the Universal Transverse Mercator (UTM) coordinate system. We also convert the translation component of the odometry observations to meters for compatibility with the UTM coordinate frame.

#### 3.1.1. Graph Construction

A pose graph is used to represent the camera poses and the motion constraints between the poses. A node, or vertex, in the graph denotes a pose, i.e., a position and orientation, and an edge denotes the relative motion constraint given by the odometry observation. We use $v_{i}$ to denote the *i*th node in the graph and $\Delta \xi_{i}$ to denote the relative motion between $v_{i}$ and $v_{i+1}$. We construct the initial pose graph by sequentially transforming poses with the odometry observations. First, we fix the first node in the graph to the zero vector, then each subsequent pose is computed by right multiplying the previous pose with a homogeneous transformation matrix T(Δξ)∈SE(2), representing the exponential map of the odometry observation,
(28a)$$\begin{array}{cc} v_{1} & ={\left(0,0,0\right)}^{T}\;, \end{array}$$
(28b)$$\begin{array}{cc} v_{i+1} & =T\left(\xi_{i}\right)v_{i}\;,\;\forall i=\left\{1,2,\ldots ,n\right\}\;. \end{array}$$To simplify the notation, we use $\xi_{ij}$ to denote the motion constraint between the poses $v_{i}$ and $v_{j}$ where $v_{j}\overset{\mathrm{def}}{=}v_{i+1}$. We also take *n* to be equal to the number of poses, which is the number of odometry constraints plus one, i.e., $n\overset{\mathrm{def}}{=}m+1$. Associated with each motion constraint is a 3×3 covariance matrix $\Sigma_{ij}$ that represents the uncertainty of the motion. As we describe in the optimization section that follows, we use the information matrix $Q_{ij}=\Sigma_{ij}^{-1}$ to represent the strength of the edge, or constraint, in the graph.

Let C be an ordered set of 2-tuples representing the correspondences between poses and GPS readings. Thus, the tuple (i,k)∈C specifies that pose $v_{i}$ corresponds to the GPS reading $g_{k}$. We insert GPS coordinates as nodes in the graph and add an edge to the corresponding camera pose. We also translate all GPS coordinates according to the first correspondence in C. We do this by storing the translation, $g_{0}\leftarrow g_{k\in C_{1}}$, where $C_{1}$ is the first GPS-odometry correspondence, and subtracting $g_{0}$ from all coordinates in the track,
(29)$$\begin{array}{cc} g_{k} & \leftarrow g_{k}-g_{0}\;\;\forall k=\left\{1,2,\ldots ,|C|\right\}\;. \end{array}$$We store the translation so that we can invert the pose graph back to the original UTM coordinates after optimization. Associated with each GPS coordinate is a 2×2 covariance matrix representing the expected accuracy of the receiver. We invert to covariance matrix, as we did with the odometry covariance, to obtain an information matrix $Q_{k}$. The values in this matrix depend on the expected accuracy of the GPS receiver and the environment in which the receiver is operating. For example, a consumer-grade GPS device operating under a dense forest canopy will have a relatively large uncertainty and thus small values in the information matrix.

Although we anchor the GPS track to a camera pose in the graph, their global orientation will likely differ. Thus, we introduce a global orientation parameter ϕ as another node in the graph that imposes a constraint on each node representing a camera pose that has a corresponding GPS reading. See Figure 3 for an illustration of the graph.

#### 3.1.2. Optimization

Given the graph structure outline above, we seek to find the optimal configuration of the state vector,
(30)$$\begin{array}{c} s={\left(\phi ,v_{1}^{T},v_{2}^{T},\ldots ,v_{n}^{T}\right)}^{T}\;, \end{array}$$
that minimizes the sum of squared errors. The state is simply a vector consisting of the global orientation parameter followed by the camera poses, where each pose is parameterized as $v_{i}={\left(x,y,\theta \right)}^{T}$. Thus, the size of this vector is 1+dn, where *d* is the number of dimensions used to describe a camera pose, in this case 3, and *n* is the number of camera poses. The errors associated with the state configuration are given by two functions: one corresponding to the relative motion constraints given by the odometry observations and one corresponding to the constraints imposed by the observations from the GPS. The odometry error function, which is equivalent to the error function used in [49], is given by
(31)$$\begin{array}{cc} r_{ij}\left(v_{i},v_{j}\right) & =T{\left(\xi_{ij}\right)}^{-1}\left(T{\left(v_{i}\right)}^{-1}T\left(v_{j}\right)\right)\;,\\ \; & =\left(\begin{array}{c} R{\left(\theta_{ij}\right)}^{T}\left(R{\left(\theta_{i}\right)}^{T}\left(t_{j}-t_{i}\right)-t_{ij}\right)\\ \theta_{j}-\theta_{i}-\theta_{ij} \end{array}\right)\;. \end{array}$$
where the notation R(θ) represents a 2D rotation matrix of the form
(32)$$\begin{array}{c} R\left(\theta \right)=\left(\begin{array}{cc} \cos\theta & -\sin\theta \\ \sin\theta & \cos\theta \end{array}\right)\;, \end{array}$$
and $\theta_{i},\theta_{j}$ and $\theta_{ij}$ are the rotation angles corresponding to $v_{i},v_{j}$ and $\xi_{ij}$, respectively. The notations $t_{i},t_{j}$ and $t_{ij}$ represent the translation vectors from $v_{i},v_{j}$ and $\xi_{ij}$, respectively. The error function defined in Equation (Equation 31) gives the translational and rotational errors by first transforming pose $v_{j}$ into the coordinate frame of $v_{i}$, then computing the differences according to the odometry observation. Thus, this function returns zero when the state vector is configured according to Equations ([Disp-formula FD28a-sensors-23-07043]) and ([Disp-formula FD28b-sensors-23-07043]).

The second error function calculates the position difference between a pose and its corresponding GPS coordinate according to the global orientation parameter,
(33)$$\begin{array}{c} r_{ik}\left(v_{i},\phi \right)=R\left(\phi \right)t_{i}-g_{k}\;. \end{array}$$This function rotates the translation vector of pose $v_{i}$ about the origin of the coordinate frame according to a rotation matrix constructed from the global orientation parameter ϕ and returns the offset to the corresponding GPS coordinate.

Given the two error functions, we can write the sum of squared errors of the state configuration s as
(34)$$\begin{array}{c} \epsilon \left(s\right)=\sum_{ij}r_{ij}^{T}Q_{ij}r_{ij}+\sum_{(i,k)\in C}r_{ik}^{T}Q_{k}r_{ik}\;. \end{array}$$
where the error vectors $r_{ij}$ and $r_{ik}$ are weighted by the information matrices to incorporate the degree of belief given to the observations. This leads to the following objective function:(35)$$\begin{array}{c} s^{★}=\underset{s}{\mathrm{argmin}}\;\;\epsilon \left(s\right)\;. \end{array}$$This is a non-linear least-squares optimization problem that can be solved with the GN algorithm. Specifically, the error function is linearized about a current estimate of the state configuration and we iteratively solve the linear system and incrementally update the state vector. Since our initial estimate of the state vector might be far from a minimum solution, we use a dampened version of the GN algorithm called the Levenberg–Marquardt (LM) algorithm [57,58]. The linear system that is solved during each LM iteration takes the form
(36)$$\begin{array}{c} \left(H+\lambda I\right)\Delta s=-b\;, \end{array}$$
where Δs is the solution to the linear system that provides incremental improvements to the state vector in the non-linear solution space by $\bar{s}\leftarrow \bar{s}+\Delta s$, where $\bar{s}$ is the current estimate of the state configuration. The variable λ is a non-negative damping factor that, when large, forces the update to behave as the steepest descent and, when small, brings the algorithm closer to GN. We initialize λ=trace(H) and divide by two if the objective value decreases and multiply by two if the objective value increases.

To construct the Hessian H and the gradient vector b, we first take the derivatives of the error functions with respect to the state parameters evaluated at the current estimated of the state. We initialize the poses in the state vector according to Equations ([Disp-formula FD28a-sensors-23-07043]) and ([Disp-formula FD28b-sensors-23-07043]), and set the global orientation parameter to ϕ=0. The partial derivatives of the odometry error function can be written as
(37a)$$\begin{array}{cc} A_{ij} & ={\left.\frac{\partial r_{ij}}{\partial v_{i}}\right|}_{s=\bar{s}}=\left(\begin{array}{cc} -R{\left(\theta_{ij}\right)}^{T}R{\left(\theta_{i}\right)}^{T} & R{\left(\theta_{ij}\right)}^{T}\frac{\partial R{\left(\theta_{i}\right)}^{T}}{\partial \theta_{i}}\left(t_{j}-t_{i}\right)\\ 0^{T} & -1 \end{array}\right)\;, \end{array}$$
(37b)$$\begin{array}{cc} B_{ij} & ={\left.\frac{\partial r_{ij}}{\partial v_{j}}\right|}_{s=\bar{s}}=\left(\begin{array}{cc} R{\left(\theta_{ij}\right)}^{T}R{\left(\theta_{i}\right)}^{T} & 0\\ 0^{T} & 1 \end{array}\right)\;, \end{array}$$
where ∂R(θi)T∂θi in $A_{ij}$ is given by
(38)$$\begin{array}{c} \frac{\partial R{\left(\theta \right)}^{T}}{\partial \theta }=\left(\begin{array}{cc} -\sin\theta & -\cos\theta \\ \cos\theta & -\sin\theta \end{array}\right)\;. \end{array}$$The derivatives for the second error function $r_{ik}\left(v_{i},\phi \right)$ are defined as
(39a)$$\begin{array}{cc} C_{ik} & ={\left.\frac{\partial r_{ik}}{\partial \phi }\right|}_{s=\bar{s}}=\frac{\partial R{\left(\phi \right)}^{T}}{\partial \phi }t_{i}\;, \end{array}$$
(39b)$$\begin{array}{cc} D_{ik} & ={\left.\frac{\partial r_{ik}}{\partial v_{i}}\right|}_{s=\bar{s}}=\left(\begin{array}{cc} R(\phi ) & 0 \end{array}\right)\;. \end{array}$$The partial derivate of the rotation matrix w.r.t. the rotation angle in $C_{ik}$ takes the same form as presented in Equation (Equation 38). Note that we have not taken any derivatives w.r.t. the GPS coordinates, as we do not wish to reconfigure them, and thus they do not appear in the state vector. For clarity, we specify the dimensions of these matrices: both A and B are 3×3 matrices, C is a 2×1 matrix, and D is a 2×3 matrix. These derivatives lead to sparse Jacobian matrices for each of the error functions,
(40a)$$\begin{array}{cc} J_{ij} & ={\left(\begin{array}{ccccc} \ldots & \ldots & A_{ij} & B_{ij} & \ldots \end{array}\right)}_{3\times 1+3n}\;, \end{array}$$
(40b)$$\begin{array}{cc} J_{ik} & ={\left(\begin{array}{ccccc} C_{ik} & \ldots & D_{ik} & \ldots & \ldots \end{array}\right)}_{2\times 1+3n}\;. \end{array}$$We use ellipses to indicate that unspecified values in the matrix are zeros. Since the odometry part of the pose graph only has constraints between consecutive nodes, the Jacobian $J_{ij}$ will always have a contiguous 3×6 block of non-zero values corresponding to the odometry constraints between nodes *i* and *j*. Furthermore, the Jacobian $J_{ik}$ will always have non-zero values in the first 2×1 block corresponding to the global orientation parameter and a 2×3 non-zero block at the *i*th node representing the constraint between the GPS-odometry correspondences. Now that the Jacobians are specified, we obtain the Gauss–Netwon approximation to the Hessian matrices by
(41a)$$\begin{array}{cc} H_{ij} & =J_{ij}^{T}Q_{ij}J_{ij}\;, \end{array}$$
(41b)$$\begin{array}{cc} H_{ik} & =J_{ik}^{T}Q_{k}J_{ik}\;, \end{array}$$
and the gradient vectors by
(42a)$$\begin{array}{cc} b_{ij} & =J_{ij}^{T}Q_{ij}r_{ij}\;, \end{array}$$
(42b)$$\begin{array}{cc} b_{ik} & =J_{ik}^{T}Q_{k}r_{ik}\;. \end{array}$$

From an implementation standpoint, it is easier to construct the Hessian directly using our definitions for the non-zeros blocks in the Jacobians. The Hessian matrix for the *i*th pose in graph, assuming there is a corresponding GPS coordinate with the node, takes the form
(43)$$\begin{array}{c} H_{ij}+H_{ik}=\left(\begin{array}{ccccc} C_{ik}^{T}Q_{k}C_{ik} & \ldots & C_{ik}^{T}Q_{k}D_{ik} & \ldots & \\ \vdots & \ddots & \vdots & \; & \\ D_{ik}^{T}Q_{k}C_{ik} & \ldots & A_{ij}^{T}Q_{ij}A_{ij}+D_{ik}^{T}Q_{k}D_{ik} & A_{ij}^{T}Q_{ij}B_{ij} & \ldots \\ \vdots & \; & B^{T}Q_{ij}A_{ij} & B_{ij}^{T}Q_{ij}B_{ij} & \\ \; & \; & \vdots & \; & \ddots \end{array}\right)\;. \end{array}$$Since the Jacobians are sparse, the resulting Hessian matrix is also sparse. Therefore, in practice, it is advantageous to use a memory-efficient sparse storage scheme for these matrices, e.g., compressed sparse column or row matrices.

The gradient vector for the *i*th pose can be constructed directly with
(44)$$\begin{array}{c} b_{ij}+b_{ik}=\left(\begin{array}{c} C_{ik}^{T}Q_{k}r_{ik}\\ \vdots \\ A_{ij}^{T}Q_{ij}r_{ij}+D_{ik}^{T}Q_{k}r_{ik}\\ B_{ij}^{T}Q_{ij}r_{ij}\\ \vdots \end{array}\right)\;. \end{array}$$The final Hessian matrix and gradient vector for the linear system are obtained by summing over all the constraint-wise Hessians and descent vectors,
(45a)$$\begin{array}{c} H=\sum_{ij}H_{ij}+\sum_{(i,k)\in C}H_{ik}\;, \end{array}$$
(45b)$$\begin{array}{c} b=\sum_{ij}b_{ij}+\sum_{(i,k)\in C}b_{ik}\;. \end{array}$$This linearization is performed during each iteration of the LM algorithm. We take advantage of the sparse structure of the Hessian and solve the system using sparse Cholesky factorization. We terminate the algorithm when the Euclidean norm of the linear increment to the state vector ∥Δs∥ is less than some small threshold, e.g., 0.001, and take the optimal configuration as $s^{★}=\bar{s}+\Delta s$. Finally, we rotate each pose in the optimal state vector according the global orientation parameter and translate back to the UTM coordinate system and update the rotation component of each pose,
(46)$$\begin{array}{c} {\hat{v}}_{i}=\left(\begin{array}{c} R\left(\phi^{★}\right)t_{i}^{★}+g_{0}\\ \theta_{i}^{★}+\phi^{★} \end{array}\right)\;,\;\forall {\left\{i\right\}}_{1}^{n}\;. \end{array}$$Hereinafter, we omit the global orientation parameter and denote the globally aligned state vector as
(47)$$\begin{array}{c} \hat{s}={\left({\hat{v}}_{1}^{T},{\hat{v}}_{2}^{T},\ldots ,{\hat{v}}_{n}^{T}\right)}^{T}\;. \end{array}$$In order to resolve the global orientation parameter, it is required that we have a minimum of two GPS observations. We also add robustness to non-Gaussian distributed GPS errors using the same approach as in Section 2.2.2. We compute weights for the residuals corresponding to the global position coordinates using Tukey’s biweight cost function, and the weights are used to scale the information matrix $Q_{k}$.

We conclude this section by providing a note on the values in the information matrices. In general, the information matrix for the odometry observations consists of large values relative to the values in the information matrix for the global position observations. This is consistent with the fact that global position measurements are typically degraded under the canopy of a forest. Furthermore, VO is expected to perform well in feature-rich environments, such as a forest. We also note that we provide a substantially larger value to the position in the odometry information matrix corresponding to the heading. This makes the translation component of the nodes in the pose graph more elastic in order to conform to the GPS track while keeping the heading stiff to help maintain an accurate reconstruction of tree stem observations, which we address in the next section.

### 3.2. Local Refinement

Given the globally aligned state vector $\hat{s}$, we will now optimize for a refined state vector by taking into account observations of tree stems. There are three main steps in local refinement: (1) We transform each tree stem observation to the world coordinate frame according to the globally aligned state vector we optimized in the previous section. (2) We associate the observations corresponding to an individual tree stem position in the global map. (3) We optimize for a new configuration of the state that minimizes both the error functions from the previous section and an additional error function representing the discrepancy between observations of tree stems and their associated global position.

#### 3.2.1. Graph Augmentation

Tree stems are detected in each frame using the convolutional neural network (CNN) object detector outlined in [59]. The input image to the network is resized to a resolution of 128 columns and 352 rows for real-time performance. We also detect the ground plane and breast height using the RANSAC-based algorithm presented in [60]. For each bounding box predicted by the CNN object detector, we extract the image coordinate, where the center-line of the bounding box along the *u*-axis of the image intersects the ground plane positioned at breast height. A disparity value is assigned to this image coordinate using Equation (Equation 5) in [59]. Using the inverse projection matrix and the disparity assignment, we project the image coordinate to $R^{3}$. We represent a tree stem observation in the camera coordinate frame of the *i*th camera pose as $z_{iq}={\left(x,y\right)}^{T}$, where *x* and *y* correspond to the back-projected image coordinate along the *x* and *z* axes of the camera coordinate frame. The index $q\in Z_{i}$ where $Z_{i}$ is a set of indices denoting the observations of stems from the *i*th camera. Thus, $|Z_{i}|$ is the number of tree stem observations in camera *i* and |Z| is the number of camera poses in the graph. We use the notation (i,q)∈Z to index the $q^{\mathrm{th}}$ observation in camera *i*.

Recalling that a camera pose in the globally aligned state vector is represented as ${\hat{v}}_{i}={\left(\hat{x},\hat{y},\hat{\theta }\right)}^{T}$, we can transform the observations from the camera coordinate frame to the world frame with
(48)$$\begin{array}{c} {\hat{z}}_{iq}=R\left({\hat{\theta }}_{i}\right)z_{iq}+{\hat{t}}_{i}\;,\;\forall \left(i,q\right)\in Z\;. \end{array}$$Given all tree stem observations in the global coordinate frame, we perform data association by clustering spatially similar observations. We use the density-based spatial clustering of applications with the noise (DBSCAN) algorithm [61] to cluster the observations. DBSCAN takes two parameters and a distance function. For the distance function, we simply use the Euclidean distance. The two parameters correspond to the search radius and the minimum number of points required for a cluster. We use a search radius of 1 m and 10 as the minimum number of points in a cluster. The algorithm yields a label for each observation that specifies to which cluster observation ${\hat{z}}_{iq}$ belongs. A label equal to zero denotes an outlier, i.e., an observation that does not belong to any cluster. We denote the set of unique labels as L and use the correspondence set M to specify that observation ${\hat{z}}_{iq}$ is assigned to label l∈L. The notation $\left(\left(i,q\right),l\right)\in M$ is used to denote that observation ${\hat{z}}_{iq}$ is assigned to cluster *ℓ*.

For each cluster of tree stem observations, we compute the center of the clusters by
(49)$$\begin{array}{c} m_{l}=\frac{1}{\sum 1_{l}}\sum_{(i,q)\in Z}1_{l}{\hat{z}}_{iq}\;,\;\forall l\in L\;, \end{array}$$
where $m_{l}={\left(x,y\right)}^{T}$ is the center of the cluster in the global coordinate frame, and the notation $1_{l}$ takes 1 when the observation ${\hat{z}}_{iq}$ is assigned to cluster *ℓ* and zero otherwise. Given the cluster centers we extend our state vector as
(50)$$\begin{array}{c} \hat{s}=\left({\hat{v}}_{i}^{T},{\hat{v}}_{2}^{T},\ldots ,{\hat{v}}_{n}^{T},m_{1}^{T},m_{2}^{T},\ldots ,m_{l}^{T}\right)\;. \end{array}$$Note that we omitted the global orientation parameter. We can reconcile this in the Hessian matrix and gradient vector by removing the first row and column in the Hessian and the first row in the gradient vector, and redefining $D_{ik}$ as $\left(I_{2},0\right)$. See Figure 4 for a graphical illustration of the augmented graph.

#### 3.2.2. Refinement

In order to find an optimal configuration of the new state vector, we must define an error function corresponding to the expected tree stem location that we initialized by computing the cluster centers. The error is simply the difference between the expected position of the stem $m_{l}$ and the observation ${\hat{z}}_{iq}$. Since the observations in the global coordinate frame are subject to change during optimization, we will define the error functions on the observations from the coordinate frame of the corresponding camera, i.e., $z_{iq}$, as we did with the error functions defined in the previous section. Thus, the error function is defined as
(51)$$\begin{array}{c} r_{il}\left({\hat{v}}_{i},m_{l}\right)=R{\left({\hat{\theta }}_{i}\right)}^{T}\left(m_{l}-{\hat{t}}_{i}\right)-z_{iq}\;, \end{array}$$
which leads to an additional term in the objective function,
(52)$$\begin{array}{c} \epsilon \left(\hat{s}\right)=\sum_{ij}r_{ij}^{T}Q_{ij}r_{ij}+\sum_{(i,k)\in C}r_{ik}^{T}Q_{k}r_{ik}+\sum_{((i,q),l)\in M}r_{il}^{T}Q_{iq}r_{il}\;. \end{array}$$The matrix $Q_{iq}$ is the information matrix for the $q^{\mathrm{th}}$ observation in camera *i*. Since the information matrix in the inverse of the covariance matrix, we can define $Q_{iq}$ in terms of the expected measurement noise of the sensor. As we back projected the detected stem to the camera coordinate frame using parameters from the stereo camera, the information matrix will, in general, have relatively small values for stems detected far from the camera rig.

We proceed in the same manner as we did in the previous section by linearizing the error function and constructing the Jacobian to compute the Hessian matrix and gradient vector. The partial derivatives of the new error function can be written as
(53a)$$\begin{array}{cc} E_{il} & ={\left.\frac{\partial r_{il}}{\partial {\hat{v}}_{i}}\right|}_{s=\hat{s}}=\left(\begin{array}{cc} -R({\hat{\theta }}_{i}) & \frac{\partial R{\left({\hat{\theta }}_{i}\right)}^{T}}{\partial {\hat{\theta }}_{i}}\left(m_{l}-{\hat{t}}_{i}\right) \end{array}\right)\;, \end{array}$$
(53b)$$\begin{array}{cc} F_{il} & ={\left.\frac{\partial r_{il}}{\partial m_{l}}\right|}_{s=\hat{s}}=R\left({\hat{\theta }}_{i}\right)\;. \end{array}$$$E_{il}$ is a 2×3 matrix and $F_{il}$ is a 2×2 matrix. These definitions can be used directly to compute the blocks with the sparse Hessian matrix,
(54)$$\begin{array}{c} H_{il}=\left(\begin{array}{ccccc} \ddots & \; & & \; & \\ \; & E_{il}^{T}Q_{iq}E_{il} & \ldots & E_{il}^{T}Q_{iq}F_{il} & \\ \; & \vdots & \ddots & \vdots & \\ \; & \; & & F_{il}^{T}Q_{iq}F_{il} & \\ \; & \; & & \; & \ddots \end{array}\right)\;, \end{array}$$
where the Hessian matrix is now a (1+3n+2|L|)×(1+3n+2|L|) matrix. Similarly, the entries to the gradient vector are
(55)$$\begin{array}{c} b_{il}=\left(\begin{array}{c} \vdots \\ E_{il}^{T}Q_{iq}r_{il}\\ \vdots \\ F_{il}^{T}Q_{iq}r_{il}\\ \vdots \end{array}\right) \end{array}$$The complete linear system is constructed by summing the Hessian matrices and gradient vectors for all the tree stem observations and adding them to the other Hessians and gradient vectors defined in Equation (45). Again, the first row and column of the Hessian, as well as the first row in the gradient vector, are removed to account for the omission of the global orientation parameter. We solve the linear system, $\left(H+\lambda I\right)\Delta s=-b$, using sparse Cholesky factorization and update the state vector by $\hat{s}\leftarrow \Delta s$. As we did for global alignment, we terminate the algorithm when Δs is less than some predetermined convergence threshold.

## 4. Analysis and Discussion

To test VO and the localization and mapping algorithms outlined above, we acquired a video sequence of a 1115 m path through a sparse ponderosa pine (*Pinus ponderosa* Douglas ex Lawson) forest in Western Montana. The video was captured by walking a hand-held 12 cm baseline ZED stereo camera [62] through the forest. The camera was operated at 10 frames per second and VGA resolution (480 × 640). Mounted on top of the camera field monitor was an antenna connected to a GlobalTop FGPMMOPA6H GPS module [63] that was queried for a GPS coordinate reading and a PDOP value after each video frame capture. The GPS coordinates and video frames were stored on an embedded backpack computer. Figure 5a shows the GPS coordinate readings projected on the UTM coordinate system after translating the position track by subtracting the first GPS coordinate from all coordinates.

We estimated camera egomotion using the direct VO algorithm presented in Section 2. We used a camera resolution of 240 × 320 to estimate egomotion and a 25% gradient magnitude threshold during pixel selection. Frame-to-frame egomotion parameters were composed to construct an odometry track using Equation (28). Figure 5b shows the visual odometry position track after converting the translation components of the egomotion parameters to meters. Figure 6 shows the GPS track (black line) and the optimized VO track after global alignment and local refinement (red line). The blue dashed line in Figure 6 shows the VO track after only applying the optimized global orientation parameter. As the figure suggests, the VO position track, although locally consistent, is subject to drift after approximately 300 m.

### 4.1. Localization

In this section, we demonstrate the performance of GPS and VO integrated localization under various scenarios of degraded and intermittent GPS reception. We refer to the optimized position track shown in Figure 6 (red line) as the ground truth position track. We acknowledge that this track has not actually been ground truthed with survey grade equipment; however, this track is deemed optimal, given the available data.

#### 4.1.1. Degraded GPS Reception

We simulated degraded GPS reception by adding zero mean Gaussian noise with a standard deviation of 5 m to each observed GPS coordinate. The incremental update to the state vector and the current estimate of the state converged after 15 LM iterations. Figure 7 shows six snapshots during optimization. The global orientation parameter converged after the 6th iteration. Figure 8 shows the converged path after global alignment and local refinement with a root mean squared error (RMSE) of less than 0.1 m compared to the ground truth position track. This suggests that, as long as GPS errors are Gaussian distributed, we can expect accurate position tracking.

We also tested robustness to non-Gaussian distributed noise in GPS readings. We randomly selected 5% of the coordinates from the GPS track and added uniformly distributed noise with a range of 0 to 200 m. Figure 9 shows the converged path after global alignment and local refinement with non-Gaussian distributed noise in GPS coordinates plotted over the ground truth path (RMSE < 0.1 m). According to this result, the algorithm is insensitive to GPS coordinate outliers.

#### 4.1.2. Intermittent GPS Reception

When GPS is used under an extremely dense forest canopy, reception might only be intermittently reliable when the receiver is stationary for long periods of time or when the receiver crosses openings in the canopy. We simulated this scenario by extracting 12 coordinates from the GPS track; we extracted the starting position, ending position and randomly selected 10 coordinates along 100 m intervals from the track. Figure 10 shows six snapshots during optimization with 12 intermittent GPS coordinate readings. The algorithm converged after 21 iterations. Following local refinement, the optimized path had a RMSE of 2.7 m compared to the ground truth position track in Figure 11. This is an important application of the proposed algorithm in situations where GPS reception is only available intermittently.

### 4.2. Mapping

In order to generate a stem map, we performed global alignment using all the coordinates from the GPS track and the VO path shown in Figure 5a,b. Figure 12 shows the result after clustering the observed tree stem position. The black dots in the figure denote the cluster centers, and the ellipses around each center show the covariance matrix of the clusters at four standard deviations. There was a total of 6205 tree stem observations and 140 clusters, i.e., individual tree stems. Figure 13 shows the optimized cluster centers, i.e., tree stem positions, and the covariance structure of the observations at four standard deviations. Local refinement converged after nine iterations. Note that the ellipses representing the covariances of the clusters are smaller after local refinement. This is a result of simultaneously optimizing the position track and the cluster centers.

To test the accuracy of the stem map generated from the local refinement step, we collected a ground truth stem map of the 12 acre forest, from which we acquired the video and GPS track. We obtained global coordinates for each individual tree in the stand using a TruePulse 360B laser range finder and a 13-bit BEI industrial rotary encoder mounted on a tripod. We installed a reflector target near the center of the stand, at which a GPS coordinate was acquired using a mapping grade receiver. We mapped subsections of the stand by first aligning the laser and encoder to north, using the appropriate declination for the area, then recorded the distance and angle to the target to globally localize the plot center. We recorded the distance and angle to each individual tree within view from the plot center. We repeated this process moving clockwise around the reflector target mapping small subsections of the stand until the entire stand was mapped.

Figure 14 shows the ground truth stem map (green triangles), the predicted tree stem position from the camera (red circles), and the optimized position track (yellow line) superimposed on an aerial photograph of the stand. We manually associated each observed tree stem position from the camera with its corresponding ground truth tree stem. Among the 140 predicted tree stem positions, we classified 2 observations as false positives, i.e., predicted tree stems that correspond to a tree in the aerial photograph that were not recorded during the collection of the ground truth stem map, and 7 tree stems as duplicate observations, i.e., observed tree stems that belong to the same ground truth stem. For all predicted and ground truth stem correspondences (n=140), we calculated a RMSE of 2.16 m. This is a 46% improvement over the best-case expected accuracy of a consumer-grade GPS device under forest canopy (4 m).

## 5. Conclusions

In this paper, we presented a real-time algorithm for VO estimation and a novel method for integrating VO and a vision-based tree stem detection system with GNSS positioning to generate accurate stem maps. Our approach is based on existing and widely applied optimization techniques, i.e., GN and LM non-linear least squares, which provide efficient solution procedures to the optimization problems formulated in this paper.

Although GNSS positioning is used to maintain a globally consistent position track, the heading of the camera is subject to drift; global positioning cannot be used to infer the heading of the camera. This issue did not appear to be significant in our dataset; however, we expect that camera heading will eventually drift since it is unconstrained and only estimated from visual egomotion. This issue can be resolved by incorporating a global direction sensor, i.e., magnetometer, to maintain a globally consistent heading. The inclusion of directional data in the global alignment optimization step requires minimal modifications to the presented algorithm. Since electronic direction sensors are relatively inexpensive, we recommend using such a sensor for large-scale mapping applications.

The successful integration of real-time localization and mapping into forestry practices presents new opportunities for enhancing decision-making processes and implementing complex silvicultural prescriptions. This research provides a practical, low-cost solution that addresses the need for accurate and efficient positioning and mapping in forested environments.

## 6. Patents

US Patent No. US011481972B2: Method of performing dendrometry and forest mapping.

## Figures and Tables

**Figure 1 sensors-23-07043-f001:**
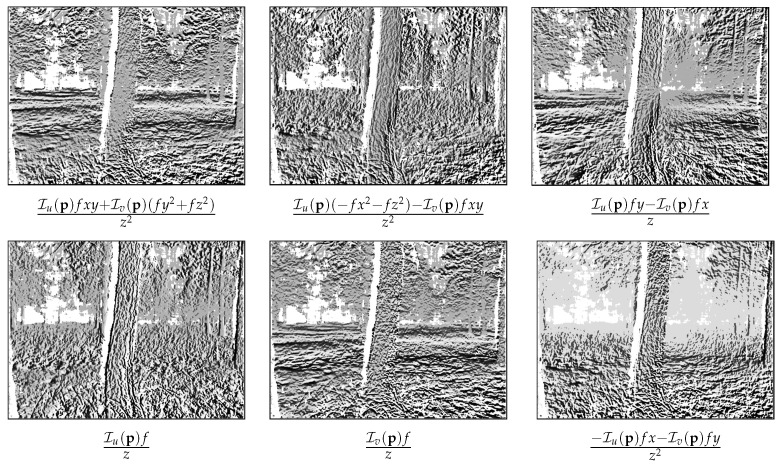
Steepest descent images; each image corresponds to a column in the Jacobian matrix.

**Figure 2 sensors-23-07043-f002:**
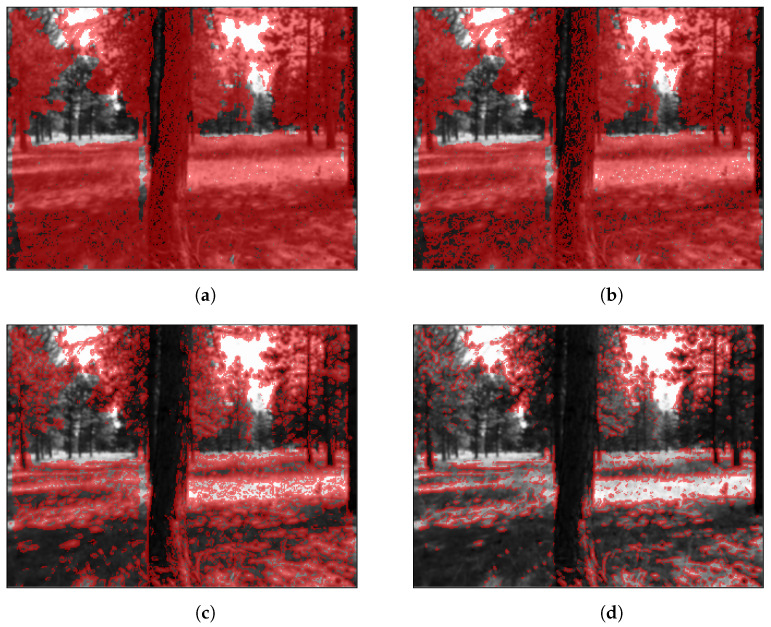
Gradient magnitude-based pixel selection; selected pixels shown in red. Subfigure captions indicate the number of selected pixels for 240 × 320 resolution images. (**a**) Dense (≈65,000 px); (**b**) 75% (≈57,600 px); (**c**) 50% (≈38,400 px); (**d**) 25% (≈19,200 px).

**Figure 3 sensors-23-07043-f003:**
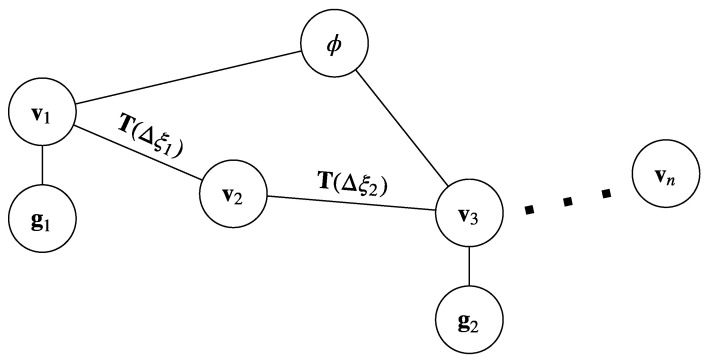
Graphical model of global alignment. Circles denote nodes (vertices) and lines denotes edges (constraints).

**Figure 4 sensors-23-07043-f004:**
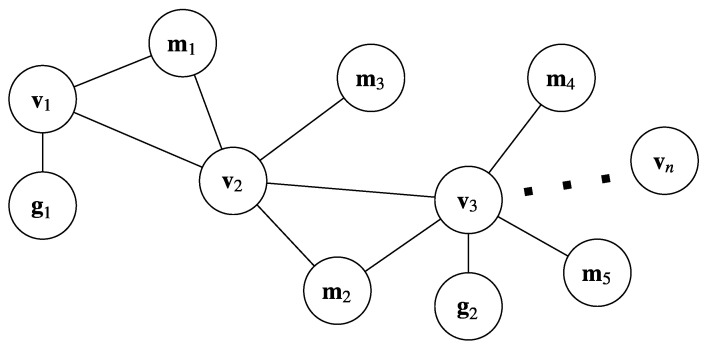
Graphical model of local refinement. Circles denote nodes (vertices) and lines denote edges (constraints). Odometry nodes are represented as v, GPS nodes as g, and tree stem positions as m.

**Figure 5 sensors-23-07043-f005:**
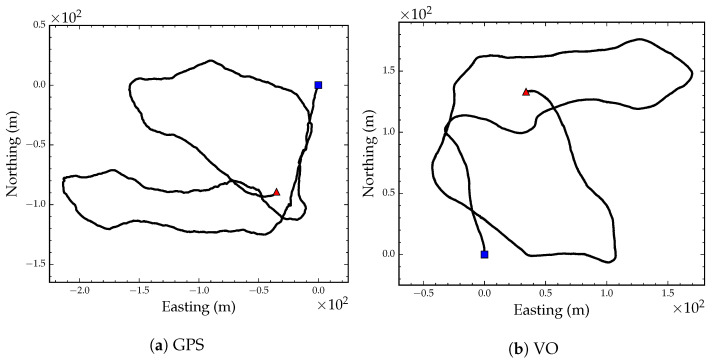
GPS position track projected on a UTM coordinate system (**a**). VO position track (**b**). The blue square denotes the starting position and the red triangle denotes the end position.

**Figure 6 sensors-23-07043-f006:**
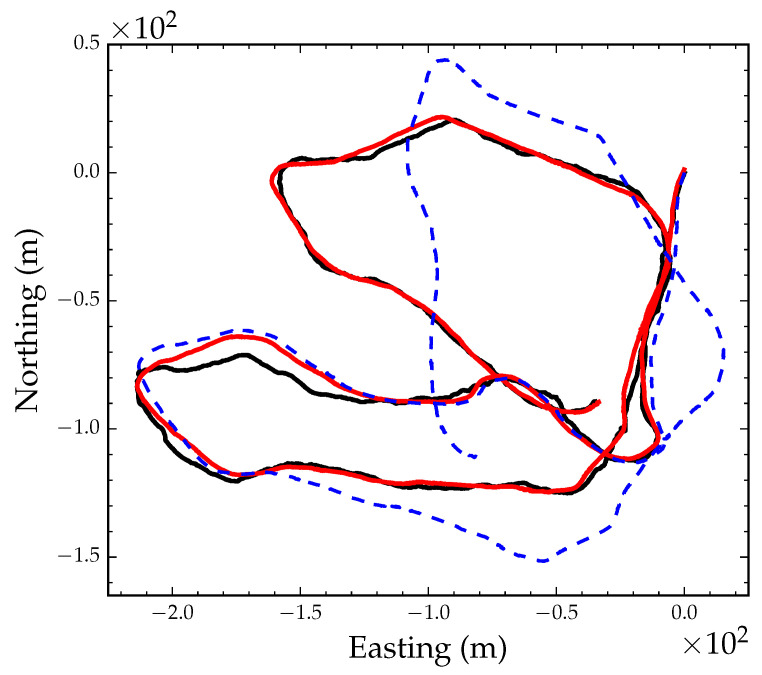
GPS position track denoted by the black solid line. Optimized position track denoted by the red solid line. VO track rotated by the global orientation parameter denoted by the blue dashed line.

**Figure 7 sensors-23-07043-f007:**
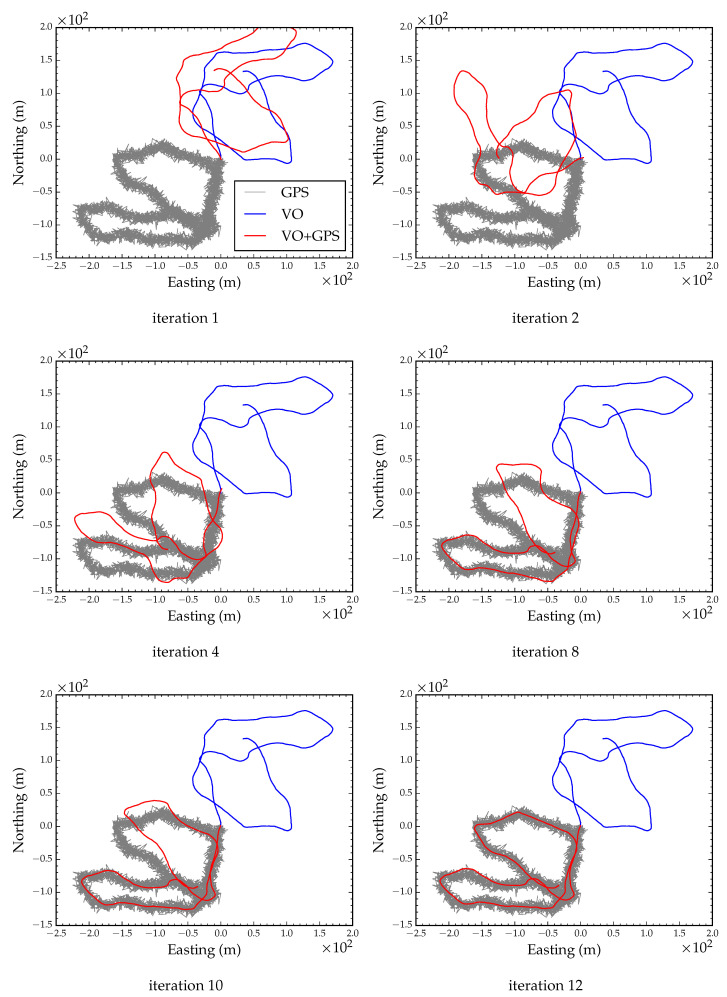
Global alignment with degraded GPS reception. Each subfigure shows a snapshot during optimization.

**Figure 8 sensors-23-07043-f008:**
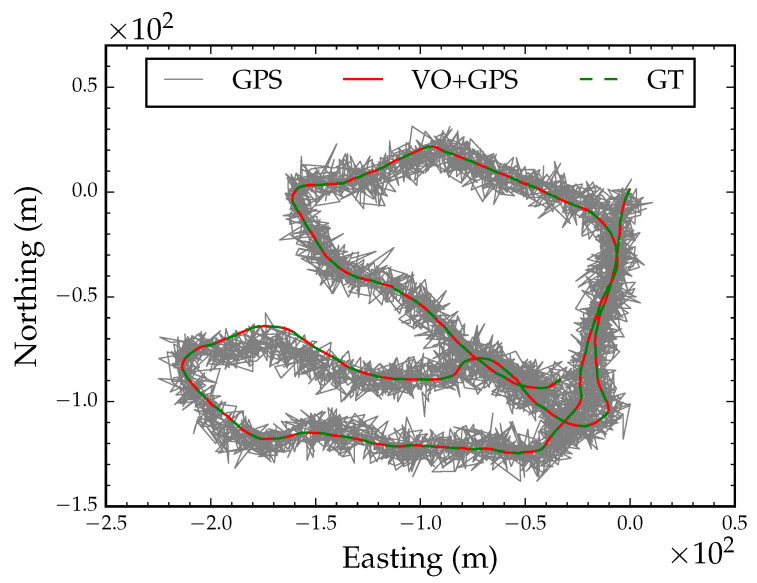
Converged globally aligned path with degraded GPS reception.

**Figure 9 sensors-23-07043-f009:**
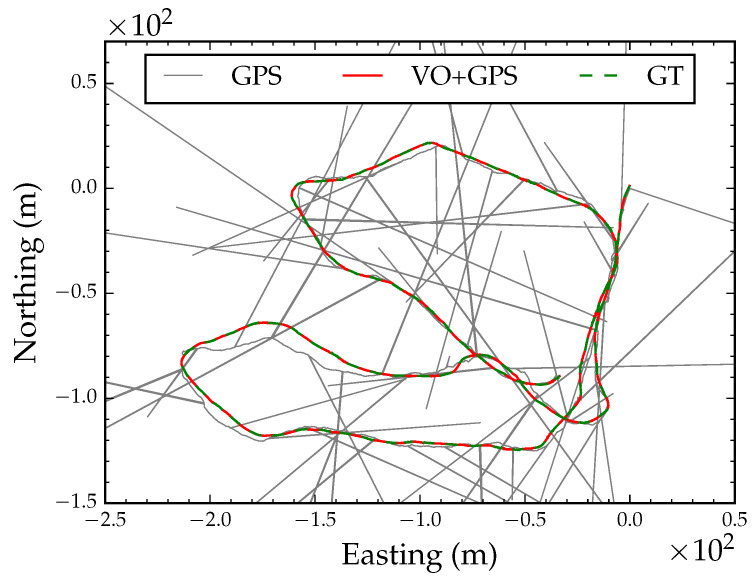
Converged globally aligned path with non-Gaussian GPS errors.

**Figure 10 sensors-23-07043-f010:**
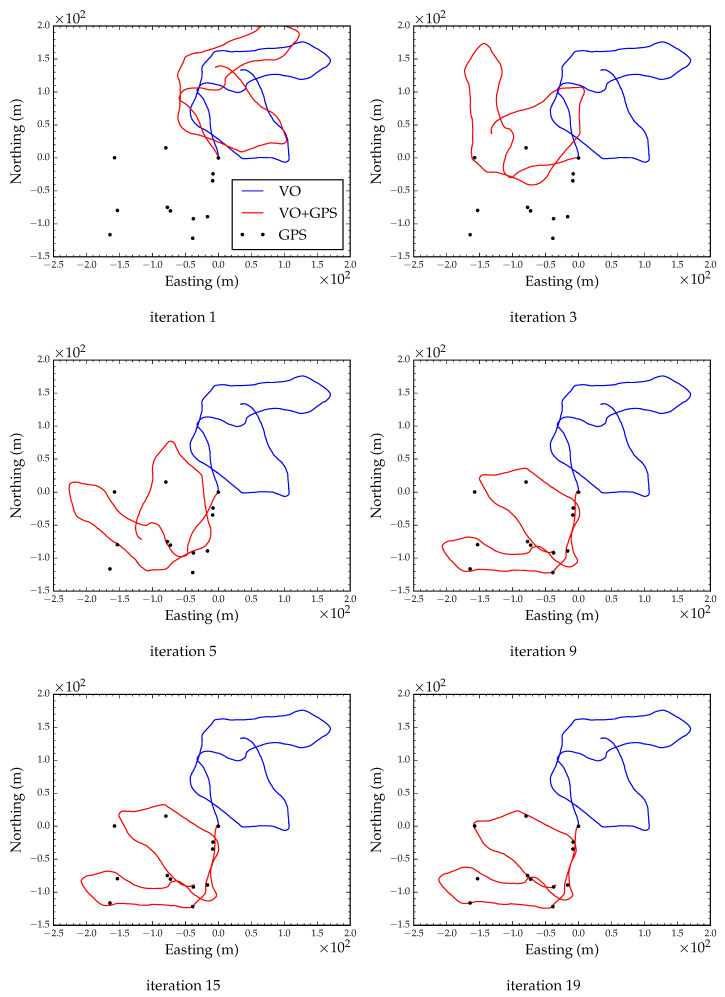
Global alignment with intermittent GPS reception. Each subfigure is a snapshot during optimization.

**Figure 11 sensors-23-07043-f011:**
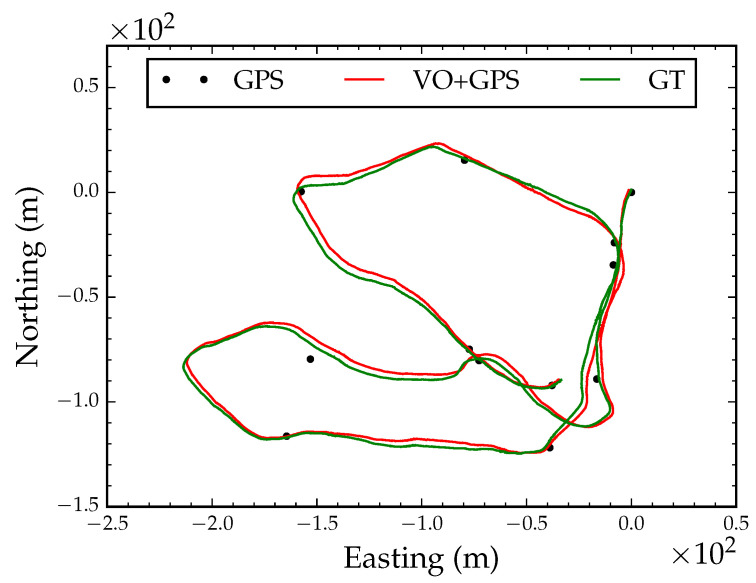
Converged globally aligned path with intermittent GPS reception.

**Figure 12 sensors-23-07043-f012:**
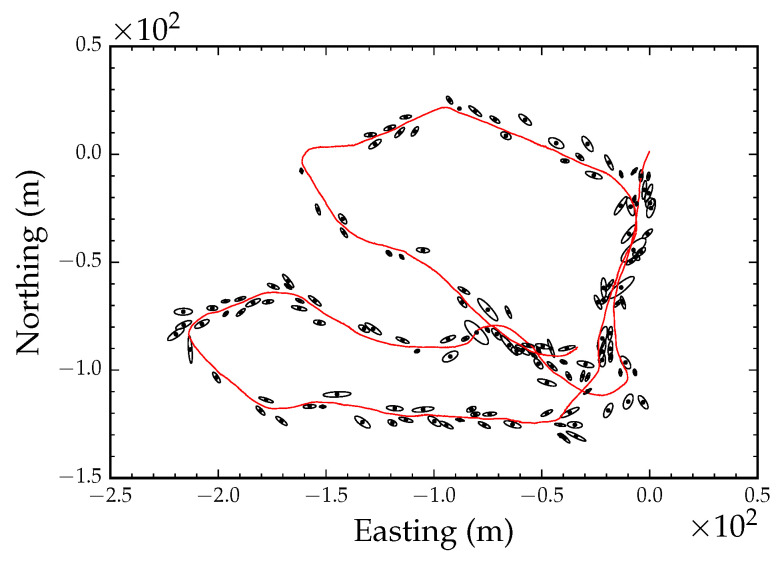
Tree stem position averages (black dots), tree stem position covariances (black ellipses), and pose track (red line) before local refinement.

**Figure 13 sensors-23-07043-f013:**
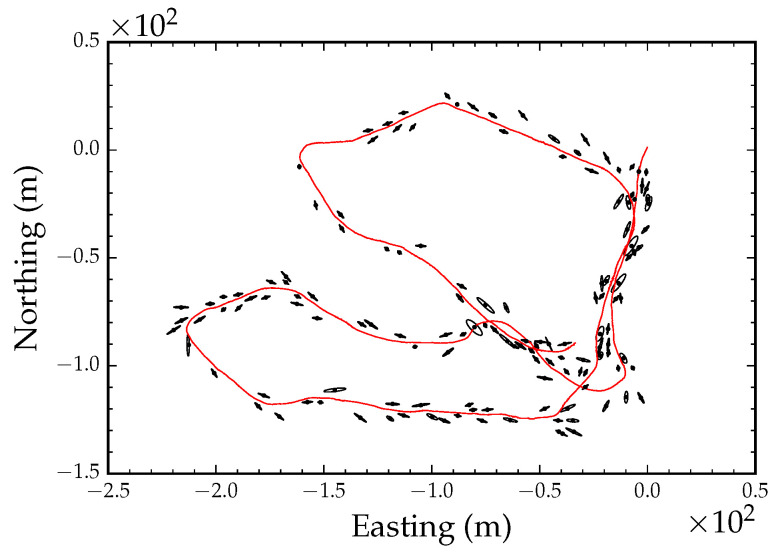
Tree stem position averages (black dots), tree stem position covariances (black ellipses), and pose track (red line) after local refinement.

**Figure 14 sensors-23-07043-f014:**
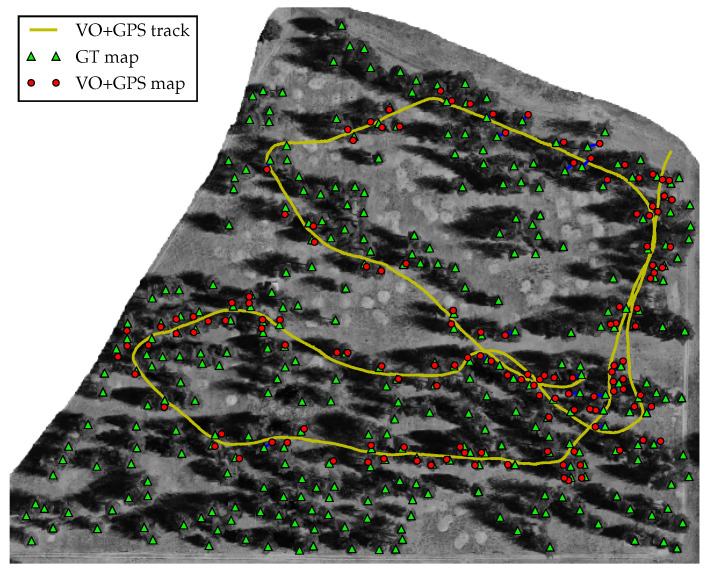
Ground truth tree stem positions and estimated tree stems positions superimposed on an aerial photograph of the 12 ac forest stand. Green triangles denote ground truth stem positions, red circles show the estimated stem position from the camera, the yellow line denotes the optimized position track, and the blue lines show association between observed and estimated tree positions.

## Data Availability

Not applicable.

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
