# Peer review of "Vision-Aided Localization and Mapping in Forested Environments Using Stereo Images"

_sensors, 2023, doi:10.3390/s23167043_

Round 1

Reviewer 1 Report

"Vision-aided localization and mapping in forested environments using stereo images" has been reviewed. The results of the evaluation are shared below:

- The novelty in this study should be given more clearly in the introduction part of the study. Especially while many studies have been done on this subject, it should be revealed which gap this study will fill.

- Although Chapter 3 contains very valuable information, it is appropriate to make it shortened.

- In the application part: in order to make more realistic and statistical comparisons, it would be more appropriate to determine the reference coordinates of the trajectory (perhaps with the classical terrestrial method).

Reviewer 2 Report

1.       According to this paper ;authors introduce visual processing algorithms that facilitate precise real-time  machine positioning and forest mapping.Authosrs employ visual egomotion estimation from a  stereo camera to maintain the position of a forestry machine during instances of degraded  or failed reception from a global navigation satellite system (GNSS)

2.       I consider the topic is  original and  relevant in the field. The study  demonstrate how precise position maintenance can be utilized to generate tree maps when  combined with a tree detection algorithm.

3.       In this paper They  introduce the concepts of visual odometry (VO) and simultaneous localization and mapping (SLAM).

4.       authors  shuold consider

a)       What is 3D egomotion. explain egomotion in detail

b)      Why do you need Image warping?

c)       What is rotation papameters and translation parameters. Please explain it with figüre in section 2.1

d)      Please In your article, write a title called Methodology and briefly explain the method used.than then explain the details

5.       I would like to recommend some new papers about your study

Avcı, C. , Budak, M. , Yağmur, N. & Balçık, F. (2023). Comparison between random forest and support vector machine algorithms for LULC classification . International Journal of Engineering and Geosciences , 8 (1) , 1-10 . DOI: 10.26833/ijeg.987605

Yiğit, A. Y., Hamal, S. N. G., Yakar, M., & Ulvi, A. (2023). Investigation and Implementation of New Technology Wearable Mobile Laser Scanning (WMLS) in Transition to an Intelligent Geospatial Cadastral Information System. SUSTAINABILITY, 15(9), 7159.

Selim, S. , Demir, N. & Oy Şahin, S. (2022). Automatic detection of forest trees from digital surface models derived by aerial images . International Journal of Engineering and Geosciences , 7 (3) , 208-213 . DOI: 10.26833/ijeg.908004

Abdurahman Yasin Yiğit, Seda Nur Gamze Hamal, Ali Ulvi & Murat Yakar (2023) Comparative analysis of mobile laser scanning and terrestrial laser scanning for the indoor mapping, Building Research &Information, DOI: 10.1080/09613218.2023.2227900

Başaran, N. , Küçük Matcı, D. & Avdan, U. (2022). Using multiple linear regression to analyze changes in forest area: the case study of Akdeniz Region . International Journal of Engineering and Geosciences , 7 (3) , 247-263 . DOI: 10.26833/ijeg.976418

Reviewer 3 Report

A demo video is strongly suggested to highlight the contributions. 
